# Bioenergetic and Cytokine Profiling May Help to Rescue More DCD Livers for Transplantation

**DOI:** 10.3390/ijms24119536

**Published:** 2023-05-31

**Authors:** Julia Hofmann, Andras T. Meszaros, Madita L. Buch, Florian Nardin, Verena Hackl, Carola J. Strolz, Bettina Zelger, Margot Fodor, Benno Cardini, Rupert Oberhuber, Thomas Resch, Annemarie Weissenbacher, Jakob Troppmair, Stefan Schneeberger, Theresa Hautz

**Affiliations:** 1organLife Organ Regeneration Center of Excellence and Daniel Swarovski Laboratory, Department of Visceral, Transplant and Thoracic Surgery, Medical University of Innsbruck, 6020 Innsbruck, Austria; julia.hofmann@i-med.ac.at (J.H.); andras.meszaros@i-med.ac.at (A.T.M.); madita.buch@i-med.ac.at (M.L.B.); florian.nardin@i-med.ac.at (F.N.); verena.hackl@i-med.ac.at (V.H.); carola.strolz@i-med.ac.at (C.J.S.); margot.fodor@i-med.ac.at (M.F.); benno.cardini@i-med.ac.at (B.C.); rupert.oberhuber@i-med.ac.at (R.O.); thomas.resch@i-med.ac.at (T.R.); annemarie.weissenbacher@i-med.ac.at (A.W.); jakob.troppmair@i-med.ac.at (J.T.); stefan.schneeberger@i-med.ac.at (S.S.); 2Department of Pathology, Medical University of Innsbruck, 6020 Innsbruck, Austria; bettina.zelger@i-med.ac.at

**Keywords:** liver transplantation, normothermic machine perfusion, mitochondria, high-resolution respirometry

## Abstract

The majority of organs used for liver transplantation come from brain-dead donors (DBD). In order to overcome the organ shortage, increasingly donation after circulatory death (DCD) organs are also considered. Since normothermic machine perfusion (NMP) restores metabolic activity and allows for in-depth assessment of organ quality and function prior to transplantation, such organs may benefit from NMP. We herein compare the bioenergetic performance through a comprehensive evaluation of mitochondria by high-resolution respirometry in tissue biopsies and the inflammatory response in DBD and DCD livers during NMP. While livers were indistinguishable by perfusate biomarker assessment and histology, our findings revealed a greater impairment of mitochondrial function in DCD livers after static cold storage compared to DBD livers. During subsequent NMPs, DCD organs recovered and eventually showed a similar performance as DBD livers. Cytokine expression analysis showed no differences in the early phase of NMP, while towards the end of NMP, significantly elevated levels of IL-1β, IL-5 and IL-6 were found in the perfusate of DCD livers. Based on our results, we find it worthwhile to reconsider more DCD organs for transplantation to further extend the donor pool. Therefore, donor organ quality criteria must be developed, which may include an assessment of bioenergetic function and cytokine quantification.

## 1. Introduction

For patients suffering from end-stage liver disease, liver transplantation (LT) remains the only definitive treatment option. However, the global demand for donor organs greatly exceeds the number of available organs, leading to a significant organ shortage [1,2]. The majority of grafts utilized for LT stem from donations after brain death (DBD). In order to overcome the organ shortage, livers from donors after circulatory death (DCD) are increasingly considered for LT. However, the acceptance rate for DCD organs is still lower compared to DBDs [3,4], which may be due to several reasons: A major risk in DCD livers is the warm ischemic phase during organ procurement, which increases the susceptibility to damage occurring during cold ischemia and reperfusion. Thus, DCDs have a higher probability of suffering from severe ischemia-reperfusion injury (IRI) [5]. Consequently, this donor type is more often associated with the risk of developing early allograft dysfunction (EAD) or biliary complications [6,7]. On the other hand, it has been shown that patient and graft survival over 1, 3 and 5 years do not significantly differ between DBD and DCD grafts. This may be attributed to the cautious selection of DCD organs and recipients, as younger donors and recipients with lower model for end-stage liver disease (MELD) scores were recruited for these studies [8,9,10,11]. In addition, the implementation of in situ normothermic regional perfusion (NRP) and normothermic machine perfusion (NMP) have gained acceptance as an improved preservation strategy for DCD organs over the past years [12,13]. A major advantage of NMP is the ability to test for organ viability and function prior to transplantation, as metabolic activity is retained under normothermic conditions [2,14]. Thus, reliable biomarkers with predictive capacity towards the clinical outcomes are required. This highlights the need for in-depth characterization and comparison of DBD and DCD organs, which could aid in the selection process of whether to transplant an organ or not.

Mitochondria are central for providing the energy demand and, hence, maintaining cellular functions. However, ischemia and subsequent reperfusion of organs during transplantation are known to induce mitochondrial injury [15,16]. Thus, the analysis of mitochondrial function has been proposed as a biomarker for organ function. Flavin mononucleotide (FMN), an indicator of mitochondrial damage, could predict clinical outcomes in liver hypothermic oxygenated machine perfusion (HOPE) [17,18]. Our group recently demonstrated the predictive value of mitochondrial respiration assessed by high-resolution respirometry (HRR) in liver NMP [19]. Further to mitochondrial impairment, cytokine secretion triggering an inflammatory response is known to contribute to IRI. When investigating immune cell dynamics during human liver NMP, our team also observed donor-type-specific patterns of cytokine levels and dynamics in the perfusate [20]. 

In the present study, we compared the overall performance of a total of 40 human DBD and DCD livers that were subjected to NMP. Specifically, we investigated whether the mitochondrial function and inflammatory activation of DCD livers differ from DBD organs during a maximum of 24 h of NMP. Herein, we demonstrate that the bioenergetic function of DCD livers recovered during NMP and eventually aligned with DBD livers, which was accompanied by a specific cytokine pattern. Our findings underline the potential of NMP in combination with in-depth (bioenergetic and immunologic) organ assessment as important tools to more liberally accept DCD livers for transplantation.

## 2. Results

### 2.1. Donor Demographics for DBD and DCD Livers and Normothermic Machine Perfusion

In this retrospective study, 40 livers were included, of which 27 stemmed from DBD donors and 13 from DCD donors (Figure 1). 

The indications to utilize NMP prior to LT, such as (A) complex recipients, (B) suboptimal donor organ quality and (C) logistics, are shown in Figure 2. While a combination of suboptimal organ quality and logistics was the main reason to apply NMP in DBD livers, the indication for NMP in all DCD livers was solely donor-related. 

Baseline donor demographics are summarized in Table 1. Overall, 57.5% were male and 42.5% were female donors. DCD organs were significantly younger compared to DBD organs (53 [48–59] years vs. 66 [53–57] years; median [IQR]; *p* = 0.0075), and the donor risk index (DRI) was significantly higher (2.35 [2.06–2.88] vs. 1.80 [1.65–1.80]; median [IQR], *p* = 0.0007). The major cause of death (COD) in DBD grafts was a cerebrovascular accident (70%), whereas the leading COD in DCD livers was due to circulatory reasons (38%).

The median warm ischemic time (WIT) in DCD livers was 25 min, and the cold ischemic time (CIT) was not significantly different between the groups. Interestingly, the median NMP duration was significantly longer in the DCD group (1106 [978–1294] min vs. 809 [653–1177] min; median (IQR); *p* = 0.0061) as well as the total preservation time, compared to DBD livers (Table 2). Careful assessment of these high-risk donor organs was ensured during the prolonged NMP times. Based on the aforementioned decision criteria obtained during NMP (see Materials), a total of 29 livers (72.5%) were found suitable for transplantation after NMP. Of the declined livers, five (18.5%) were DBD and six (46.1%) were DCD organs (Table 2, Figure 1). The high discard rate in the DCD group may be attributed to a liberal acceptance policy in combination with growing expertise in organ assessment and a focus on patient safety [21].

### 2.2. Recipient Characteristics and Postoperative Outcome

Recipient demographics and clinical outcomes for this study’s cohort and groups are summarized in Table 3. The median age of the recipients was 60 (50–68) years, with a median BMI of 25.8 (20.9–29.6) kg/m^2^. The MELD score at the time of transplantation was 12.0 (7.5–19.8). Recipient characteristics did not differ between the DBD and DCD groups. The overall 1-year survival rate was 86.2%. Two patients with a DBD organ died due to multiorgan failure caused by colon perforation or aspergillosis, while two recipients receiving a DCD organ died due to multiorgan failure caused by C. difficile infection or mycosis. However, for the benchmark cases, according to Muller et al., a 100% 1-year patient and graft survival was reported [22].

### 2.3. NMP Was Stable and Uneventful in Transplanted DBD and DCD Livers 

In general, the organ machine perfusions were uneventful in all transplanted livers with regular blood flows and physiological pressures, regardless of the donor type (Figure 3A–E). Bile production was present in DBD and DCD grafts but at a variable rate (Figure 3F). 

Serial perfusate samples were closely analyzed for aspartate aminotransferase (AST), alanine aminotransferase (ALT), lactate dehydrogenase (LDH) and lactate to monitor organ function. All transplanted grafts, irrespective of the donor type, cleared lactate. AST and ALT showed no significant differences between DBD and DCD livers for all time points during perfusion, nor were LDH levels significantly different (Figure 4).

### 2.4. Bioenergetic Function in Liver Biopsy Samples of Transplanted DBD and DCD Livers: DCD Livers Tend to Align with DBD Livers during NMP 

#### 2.4.1. Flux Control Ratios

HRR was employed to assess the bioenergetic function in tissue biopsies of transplanted DBD and DCD livers before (pre), at 1 h (hour) and at the end of NMP, as well as post-reperfusion. First, substrate preference between the donor types was investigated by analysis of flux control ratios (FCR) (Figure 5). In general, both donor types showed similar patterns of mitochondrial fueling substrate preference. However, the contribution of fatty acid oxidation (F-linked respiration) to the OXPHOS capacity was found to be higher in pre-NMP biopsies of DCD livers compared to DBD livers (0.25 [0.21–0.31] in DBD vs. 0.32 [0.30–0.36] in DCD, *p* = 0.0038). After the start of NMP, the proportion of F-linked respiration significantly decreased in DCD livers, reaching a level observed in DBD grafts by then. During the further course of NMP and post-reperfusion, no differences in the FCR were observed between DBD and DCD livers.

#### 2.4.2. Coupling Control Analysis

The mitochondrial coupling control efficiencies were analyzed for the succinate pathway since succinate-linked respiration has been shown to dominate in human livers and is sufficient to saturate OXPHOS [19]. While no significant differences in pre-NMP biopsies of transplanted DBD and DCD grafts were observed, there was a trend towards a higher OXPHOS capacity in DCD livers (41.34 [30.91–47.16] pmol/s∙mg in DBD vs. 49.07 [26.10–60.14] pmol/s∙mg in DCD) (Figure 6). This trend persisted in the early phase of NMP but faded with the prolongation of NMP and post-reperfusion. In line with this, elevated LEAK respiration was found pre-NMP and at 1 h of NMP in DCD livers compared to DBD livers. With the prolongation of NMP, LEAK respiration decreased significantly in DCD grafts (pre vs. end, *p* = 0.0203) and did not significantly differ from DBD livers by then. Together, these findings suggest a higher mitochondrial impairment in pre-biopsies of DCD livers compared to DBD livers; however, mitochondrial function seems to improve in DCD livers during NMP. Low cytochrome c control efficiency (0.09 [0.03–0.13] in DBD vs. 0.06 [0.04–0.14] in DCD) indicates no extensive damage in pre-biopsies to the outer mitochondrial membrane for both groups, and no change during NMP was found. Further to this, we calculated the P-L control efficiency as a measure of ATP production efficiency. In both groups, the median P-L control efficiency was approximately 0.8 at all time points. 

Moreover, we investigated the mitochondrial parameters in light of early allograft dysfunction (EAD) post-transplantation and compared them to livers with initial function (IF) for both groups (Figure 7). Despite a relatively small number of livers in the subgroups, our analysis revealed that DCD livers developing post-transplant EAD tend to show higher LEAK respiration and lower P-L control efficiency during NMP compared to DBD livers.

### 2.5. Tissue Microstructure Is Similarly Preserved in Transplanted DBD and DCD Livers

Histopathologic analysis of tissue biopsies pre-NMP did not show any differences in aspects such as necrosis, steatosis, inflammation, fibrosis and vascular changes between DBD and DCD transplanted livers (Table 4 and Figure 8). Moreover, NMP had no impact on liver histopathology over a course of more than 20 h of perfusion time in both groups. No significant histopathological changes for all aforementioned features were found at the end of NMP or post-reperfusion. 

### 2.6. Perfusate Cytokine Patterns Differ between Transplanted DBD and DCD Livers during NMP

The analysis of cytokine levels in the perfusate early (after 1 h) and at the end of NMP revealed no significant differences between transplanted DBD and DCD grafts for the anti-inflammatory cytokines IL-4, IL-12p70 and IL-13 (Figure 9A–C). In both groups, the levels of IL-4 were below the detection limit, and the levels of IL-12 p70 decreased significantly by the end of the perfusion. For the pro-inflammatory perfusate cytokine levels, no differences were observed early after start of NMP for both groups, while some group-specific changes were found at the end of NMP. In DBD livers, IL-5 and IL-18 increased significantly over perfusion time, whereas a significant increase for IL-1β, IL-5 and IL-6 was found in DCD livers (Figure 9G,I,J). Interestingly, some of the pro-inflammatory cytokines even significantly decreased during NMP, but this phenomenon was only observed in DBD livers. Of note, high variability between the grafts has been recognized for both groups.

### 2.7. Declined Organs: High Inter-Graft Variability of Bioenergetic Function in Both DBD and DCD Livers 

Based on our center criteria (see Materials), the decision to decline a liver for transplantation after NMP was made for 11 grafts (DBD = 5, DCD = 6). The detailed reasons can be found in Table 5. Except for the livers declined due to technical problems, higher levels of AST, ALT, LDH and lactate compared to transplanted livers have been found (Figure 10). Moreover, aggravated necrosis was found at the end of NMP in some DBD and DCD grafts, compared to pre-NMP samples. 

Mitochondrial respiration and FCR analysis revealed that substrate preferences did not differ between discarded and transplanted DBD and DCD livers, respectively. Although we found considerable intra-group variability for both donor types, there was a trend towards higher S-linked OXPHOS capacity and LEAK respiration in pre-NMP biopsies and after 1 h of NMP for DCD livers compared to DBD livers (Figure 11). For both donor types, grafts with a high P-L control efficiency and low cytochrome c control efficiency throughout the perfusion were identified, indicating a reasonable bioenergetic function of those livers. However, in other grafts, the mitochondrial parameters indicate impaired mitochondrial function due to low P-L control efficiency and elevated LEAK respiration and cytochrome c control efficiency. 

Also, perfusate cytokine measurements revealed high inter-graft variability for both groups. However, the trends in declining DBD and DCD livers were similar to those in transplanted livers, respectively (Table 6). While the anti-inflammatory cytokine IL-4 was below the detection limit and IL-12p70 decreased at the end of NMP, the pro-inflammatory cytokines IL-5 and IL-6 markedly increased in the perfusate during NMP of both DBD and DCD grafts.

## 3. Discussion

We herein studied the bioenergetic performance through a comprehensive evaluation of mitochondria by HRR in biopsies and the inflammatory response in DBD and DCD livers during NMP. While livers were indistinguishable by perfusate biomarker assessment and histology, our findings revealed a greater impairment of mitochondrial function in DCD livers after static cold storage compared to DBD livers. Interestingly, the bioenergetic function of DCD livers aligned with that of DBD livers during NMP. A distinct cytokine pattern was found to be increased at the end of NMP in DCD livers. Our study highlights that NMP can be performed uneventfully and stable regardless of the donor type, also in the majority of DCD grafts despite being classified as “high-risk organs”.

The discard rate for extended criteria livers, including DCD organs, remains high, which may be attributed to concerns about post-operative complications and poor function [10,11]. With novel technologies such as NMP allowing for in-depth assessment of organ function and viability prior to transplantation, those organs may be increasingly accepted for transplantation. At this point, various biomarkers have been suggested, but validation in large cohorts is pending [2,14,23].

The liver contains a large number of mitochondria, which are required to cover the energy demand of the organ. Thus, intact mitochondrial function has been suggested as an indicator of good organ function [15,23]. Our group recently reported a predictive value towards the early clinical outcome of mitochondrial function parameters assessed during NMP of human livers [19]. The efficiency of ATP production, reflected by the P-L control efficiency, and the integrity of the outer and inner mitochondrial membranes, reflected by LEAK respiration and cytochrome c control efficiency during the early NMP, correlated with the L-GrAFT score. In correspondence with this trial, we aimed to analyze mitochondrial function during liver NMP in the light of the donor types DBD and DCD. Irrespective of the donor type, in both cohorts—transplanted or discarded livers—succinate alone was able to saturate mitochondrial respiration pre- and during NMP. The subsequent in-depth analysis of the succinate-linked pathway revealed higher LEAK respiration in DCD livers pre-NMP compared to DBD livers. Interestingly, after the start of NMP LEAK, respiration decreased in DCD organs, which may indicate a partial recovery of the mitochondrial coupling. To date, there is only limited data available reporting on the bioenergetic function of DCD livers during machine perfusion (NMP and/or HOPE). In a study by Muller et al., significantly elevated levels of FMN were found in DCD liver throughout HOPE [17]. FMN is a component of the mitochondrial complex I that is released into the perfusate upon mitochondrial damage. A direct comparison to the results of our study obtained in DCD livers subjected to NMP remains difficult as we used the HRR for analysis of the bioenergetic function, which has not been performed before in HOPE DCD livers. Hence, it remains speculative at this point whether DCD liver grafts may benefit from dynamic machine perfusion, and further studies are required for a detailed comparison of mitochondrial performance between both preservation techniques—NMP and HOPE.

To date, little is known about donor-type-specific protein cytokine secretion [24]. Lee et al. report increasing pro-inflammatory cytokine expression levels during 6 h of NMP in three DBD and three DCD livers; however, they did not assess for donor type-specific differences [25]. In the present study, we found dynamics for both anti-inflammatory and pro-inflammatory cytokines during NMP in DBD and DCD grafts, but without any significant differences between both groups in the early phase of NMP. However, with prolongation of NMP, IL-1β, IL-5 and IL-6 increased, reaching significant differences in DBD vs. DCD livers eventually. This is consistent with the results of our previous study, where we found significantly elevated levels of IL-6 in DCD compared to DBD livers [20]. In contrast, anti-inflammatory cytokines IL-4, IL-13 and IL-12p70 decreased throughout NMP in the overall cohort, reaching significance in DBD and DCD livers for the latter. This indicates a decreased overall anti-inflammatory response with prolonged NMP in our study cohort. Based on our previous findings, where we reported the activation and mobilization of a great number of leukocytes into the perfusate during liver NMP [20], we hypothesize that in terms of cytokine production, the anti-inflammatory counter response may be exceeded by the pro-inflammatory response of activated immune cells.

In this study, one-third of a total of 40 NMP livers included were DCD organs. The decision for transplantation was based on conventional perfusate injury markers [21]. A significantly longer NMP time for these ‘high-risk’-classified DCD organs, reflected by a significantly higher DRI—compared to DBD livers, may indicate that the decision-making process was somewhat more complex and hence prolonged. Furthermore, we observed high variability between individual grafts in discarded DBD and DCD livers. Interestingly, some of the declined livers, especially in the DCD group, showed similar levels of mitochondrial parameters and cytokines compared to the transplanted livers. Hence, our findings support the idea of a wider utilization of DCD organs for transplantation through a comprehensive evaluation of liver quality and function beyond conventional biomarkers. 

There are obvious limitations to the present study. (i) The sample size is small, especially in sub-group analysis, and confirmation in a larger cohort would be required. (ii) There might be a selection bias since DCD donors were significantly younger and age-matched controls are lacking due to the small sample size. (iii) Perfusion times are variable; thus, end biopsies are uncontrolled, and comparisons for those analyses would also require matching controls.

Based on the findings of this trial, we find it worthwhile to consider more DCD organs for transplantation in order to further extend the donor pool. To aid in the process of selecting the optimal (DCD) organs for transplantation, donor organ quality criteria must be further developed, which may include an assessment of the bioenergetic function and cytokine expression pattern.

## 4. Material and Methods

### 4.1. Study Design

In total, 40 livers subjected to NMP were enrolled in this retrospective study. This study was approved by the Ethical Committee of the Medical University of Innsbruck (EK Nr. 1175/2018), and signed informed consent was obtained from all patients. Donor livers were applied to NMP prior to LT for one or a combination of the following reasons: (A) complex recipients, (B) suboptimal donor organ quality and (C) logistics. The WIT time for DCD donors is defined as the time from a mean arterial pressure below 50 mmHg or an arterial saturation of <80% until the start of cold perfusion. The CIT is defined as the time between the start of cold aortic perfusion and the start of NMP. For all perfusions, the Metra^®^ device (OrganOx Limited, Oxford, UK) was used according to the protocol described by Cardini et al. [21]. Perfusate samples were collected at 15 min, 1 h, 6 h, 12 h after the start of NMP and at the end of NMP. They were immediately analyzed for AST, ALT, LDH (all Roche Diagnostics GmbH, Mannheim, Germany), and lactate (Drott Medizintechnik GmbH, Wiener Neudorf, Austria). The decision to transplant or decline a liver after NMP was based on the previously described criteria [21]: (i) prompt decline of perfusate lactate levels to ≤2.5 mmol/L, (ii) physiological pH levels (7.30–7.45) without the need for repeated sodium bicarbonate addition, and (iii) perfusate levels of AST, ALT and LDH levels below 20,000 U/L. For the transplanted livers, early graft function was evaluated using the Liver Graft Assessment Following Transplantation (L-GrAFT) score [26], the MEAF score [27], and early allograft dysfunction (EAD) according to Olthoff et al. [28]. The total follow-up period was 1 year.

### 4.2. High-Resolution Respirometry

Tissue biopsies were taken prior to NMP (pre), 1 h after the start of NMP, at the end of NMP, and 1 h after reperfusion (post). They were immediately transferred into an ice-cold Histidine-Tryptophan-Ketoglutarate (HTK) solution (Custodiol^®^, Dr. Franz Köhler Chemie GmbH, Bensheim, Germany) and stored at 4 °C. HRR was performed using the Oxygraph2k (O2k, Oroboros Instruments, Innsbruck, Austria) as previously described in duplicates [19]. Tissue samples (20 mg) were dissected on a cooled plate at 4 °C and homogenized using the PBI-Shredder O2k-Set (Oroboros Instruments, Innsbruck, Austria) in 4 °C MiR05 respiration media (MiR05 Kit, Oroboros Instruments, Innsbruck, Austria, consisting of 0.5 mM EGTA, 3 mM MgCl_2_·6H_2_O, 60 mM lactobionic acid, 20 mM taurine, 10 mM KH2PO4, 20 mM HEPES, 110 mM D-sucrose, and the addition of 1 g/L essentially fatty acid-free bovine serum albumin) to obtain a tissue homogenate with a final wet mass of 1 mg/mL. 2 mL of the tissue homogenate were immediately transferred into each of the O2k chambers. The pre-defined Substrate-Uncoupler-Inhibitor Titration (SUIT) protocols as shown in Appendix A were applied. Chemicals were titrated using microsyringes (Oroboros Instruments, Innsbruck, Austria) as soon as respiration reached a steady state. Respiration rates were recorded with DatLab 7 software (DatLab 7.4, Oroboros Instruments, Innsbruck, Austria) and expressed as O2 flux per wet weight of tissue mass. 

Using the SUIT01 protocol (Appendix A), the contribution of the three different pathways—the fatty acid oxidation (F-linked respiration), NADH-linked pathway (N-linked respiration) and succinate (S-linked respiration)—within the OXPHOS capacity was evaluated. Therefore, the flux control ratio (FCR) was calculated by normalizing the respiratory capacities to an internal reference state. This allows us to assess the relative contribution of the F-linked, N-linked and S-linked respirations to the OXPHOS capacity, which is reached after the addition of the substrates for all three pathways. 

In protocol SUIT02, the coupling states LEAK (L), OXPHOS (P) and OXPHOS after cytochrome c (Pc) were evaluated, which allowed assessing the oxidative phosphorylation and its efficiency, as well as the integrity of the mitochondria. LEAK (L) respiration is measured in the presence of reducing substrates (such as succinate) but in the absence of adenosine diphosphate (ADP). Thus, oxygen consumption is assessed, which compensates for the electron leak through the inner mitochondrial membrane. Next, OXPHOS (P) is measured after the addition of ADP, and consequently, ATP is generated. As a next step, OXPHOS after cytochrome c (Pc) is measured by titration of cytochrome c, which is indicative of the integrity of the outer mitochondrial membrane. Moreover, the P-L control efficiency, which allows for evaluation of the efficiency of mitochondrial ATP production, was calculated as 1-LEAK respiration/OXPHOS (1-L/P). The calculation of the cytochrome c control efficiency as 1-OXPHOS/OXPHOS after cytochrome c (1-P/Pc) points towards inner mitochondrial membrane integrity [29]. A detailed description can also be found in our previous publication [19].

### 4.3. Histology

Tissue biopsies (pre-NMP, at the end of NMP, and post-reperfusion) were fixed in 10% buffered formalin and processed by routine procedures for paraffin sectioning. Paraffin sections of 4 µM were generated, and Haematoxylin & Eosin (H&E) staining was performed. Samples were examined by light microscopy and scored as previously described [19].

### 4.4. Cytokine Analysis

Perfusate samples were collected at 1 h and, at the end of NMP, centrifuged at 3000 rpm for 15 min. The supernatant was transferred and stored at −80 °C until measurement. The cytokines GM-CSF, IFN gamma, IL-1 beta, IL-2, IL-4, IL-5, IL-6, IL-12p70, IL-13, IL-18 and TNF alpha were analyzed using the Th1/Th2 Cytokine 11-Plex Human ProcartaPlexTM Panel (Cat# EPX110-10810-9001, Invitrogen, Waltham, MA, USA) according to the manufacturer’s instructions. 25 µL of thawed serumwas processed in 96-well plates using magnetic beads. Reference standard concentrations for each cytokine were assayed for the generation of standard curves to calculate serum cytokine concentrations.

### 4.5. Statistical Analysis

GraphPad Prism software (GraphPad Prism version 9.4.1; San Diego, CA, USA) was used for all statistical analyses and figures. *p*-values < 0.05 were considered statistically significant. The Shapiro–Wilk Test was applied to test for normal distribution. The descriptive statistics are represented as the median and interquartile range (IQR) for the non-normally distributed parameters. For parameters with repeated measurements at different time points, a repeated measures ANOVA (RM ANOVA) with Tukey’s multiple comparisons test was applied to assess differences within the group, and a two-way ANOVA with Sidak’s multiple comparisons was applied to assess differences at the different time points.

## Figures and Tables

**Figure 1 ijms-24-09536-f001:**
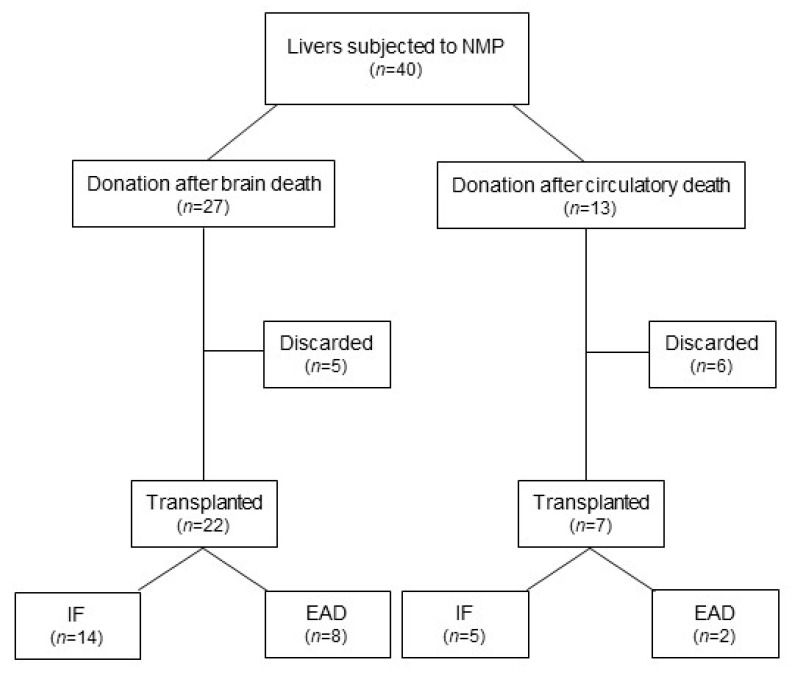
Consort diagram. NMP, normothermic machine perfusion; IF, initial function; EAD, early allograft dysfunction.

**Figure 2 ijms-24-09536-f002:**
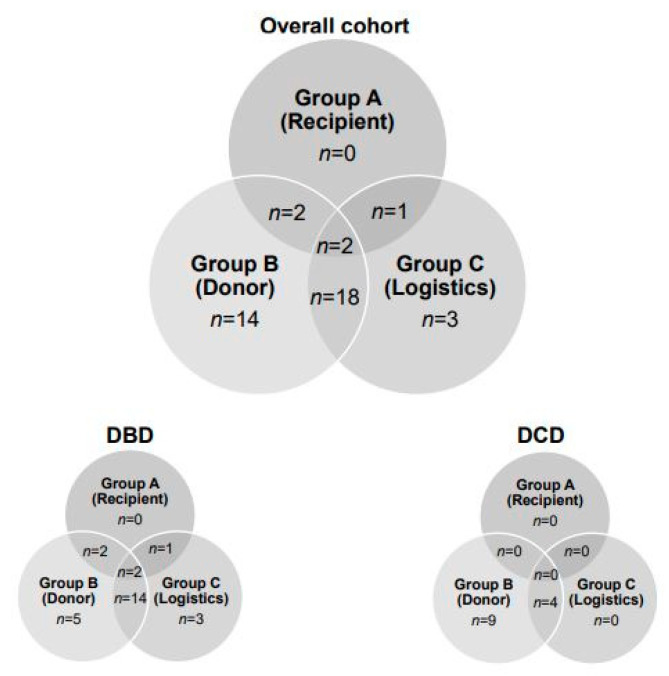
Indications for applying normothermic machine perfusion to donor’s livers. DBD, donation after brain death; DCD, donation after circulatory death.

**Figure 3 ijms-24-09536-f003:**
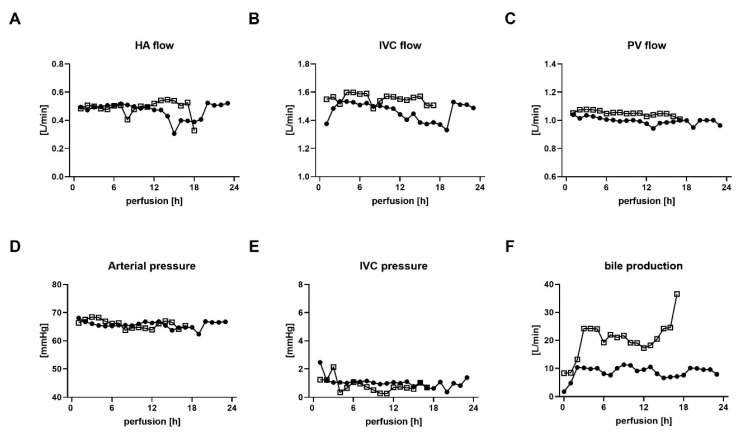
Assessment of machine perfusion parameters in transplanted DBD and DCD livers. (**A**) Hepatic artery (HA) flow, (**B**) Inferior vena cava (IVC) flow, (**C**) Portal vein (PV) flow, (**D**) Hepatic artery pressure, (**E**) Inferior vena cava (IVC) pressure and (**F**) bile production for DBD donors (solid circles) and DCD donors (open boxes). h, hours; DBD, donation after brain death; DCD, donation after circulatory death.

**Figure 4 ijms-24-09536-f004:**
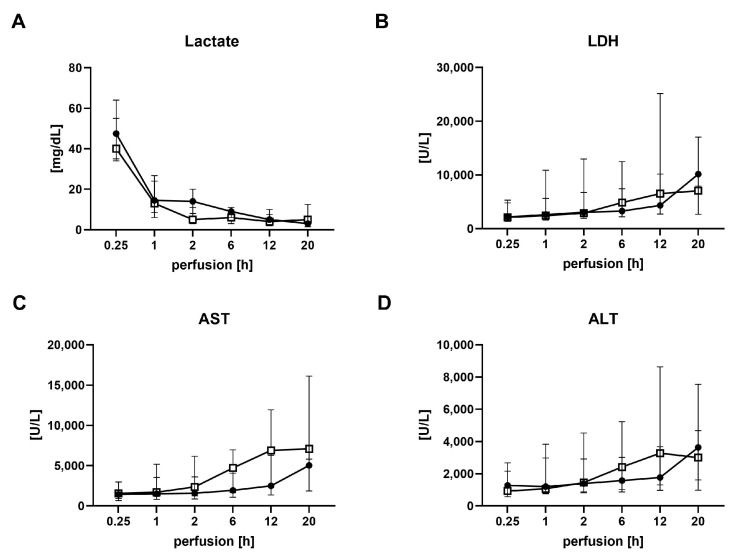
Analysis of perfusate during normothermic machine perfusion in transplanted DBD and DCD livers. (**A**) Lactate, (**B**) Lactate dehydrogenase (LDH), (**C**) Aspartate aminotransferase (AST) and (**D**) Alanine aminotransferase (ALT) for DBD donors (solid circles) and DCD donors (open boxes) with subsequent transplantation. Values are expressed as the median and interquartile range. h, hours; DBD, donation after brain death; DCD, donation after circulatory death.

**Figure 5 ijms-24-09536-f005:**
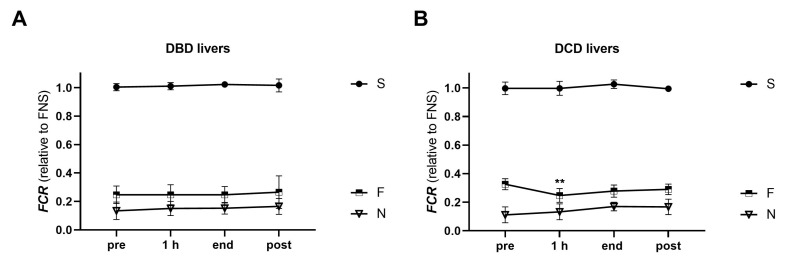
Analysis of the substrate contribution of mitochondrial respiration in tissue biopsies of transplanted DBD and DCD livers pre-NMP, at 1 h and at the end of NMP, and post-reperfusion. Flux control ratios (FCR) were calculated to assess the relative contribution of the three substrate pathways to the maximum OXPHOS capacity for DBD (**A**) and DCD (**B**) livers. Values are represented as the median and interquartile range. (** *p* < 0.05, ANOVA tests between time points and between the groups; details are described in the section on statistical analysis). NMP, normothermic machine perfusion; h, hour; DBD, donation after brain death; DCD, donation after circulatory death; S, succinate-linked respiration; F, fatty acid oxidation-linked respiration; N, NADH-linked respiration.

**Figure 6 ijms-24-09536-f006:**
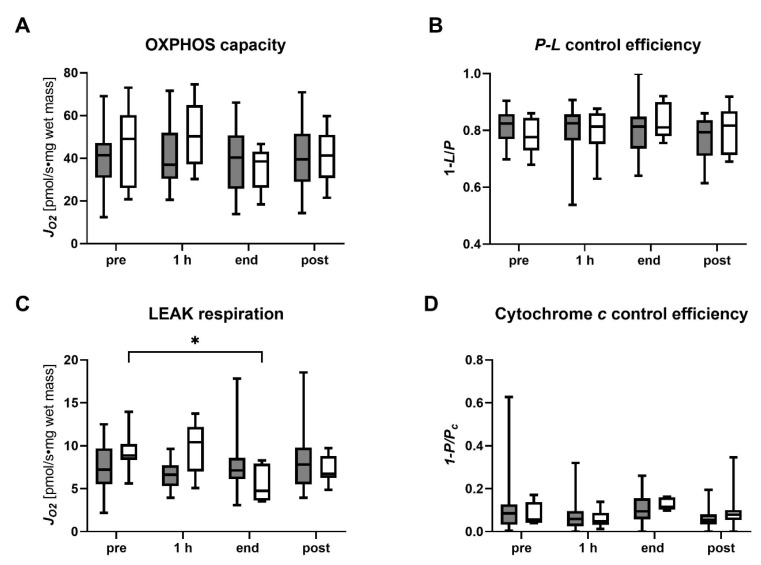
Analysis of succinate-linked mitochondrial respiration in tissue biopsies of transplanted DBD and DCD livers pre-NMP, at 1 h and at the end of NMP, and post-reperfusion. (**A**) OXPHOS capacity, (**B**) P-L control efficiency, (**C**) LEAK respiration and (**D**) cytochrome *c* control efficiency were measured and calculated following the protocol as described in Appendix A. Values are represented as the median and interquartile range for DBD (grey) and DCD (white) livers. (* *p* < 0.05, ANOVA tests between time points and between the groups; details are described in the section on statistical analysis). NMP, normothermic machine perfusion; h, hour; DBD, donation after brain death; DCD, donation after circulatory death.

**Figure 7 ijms-24-09536-f007:**
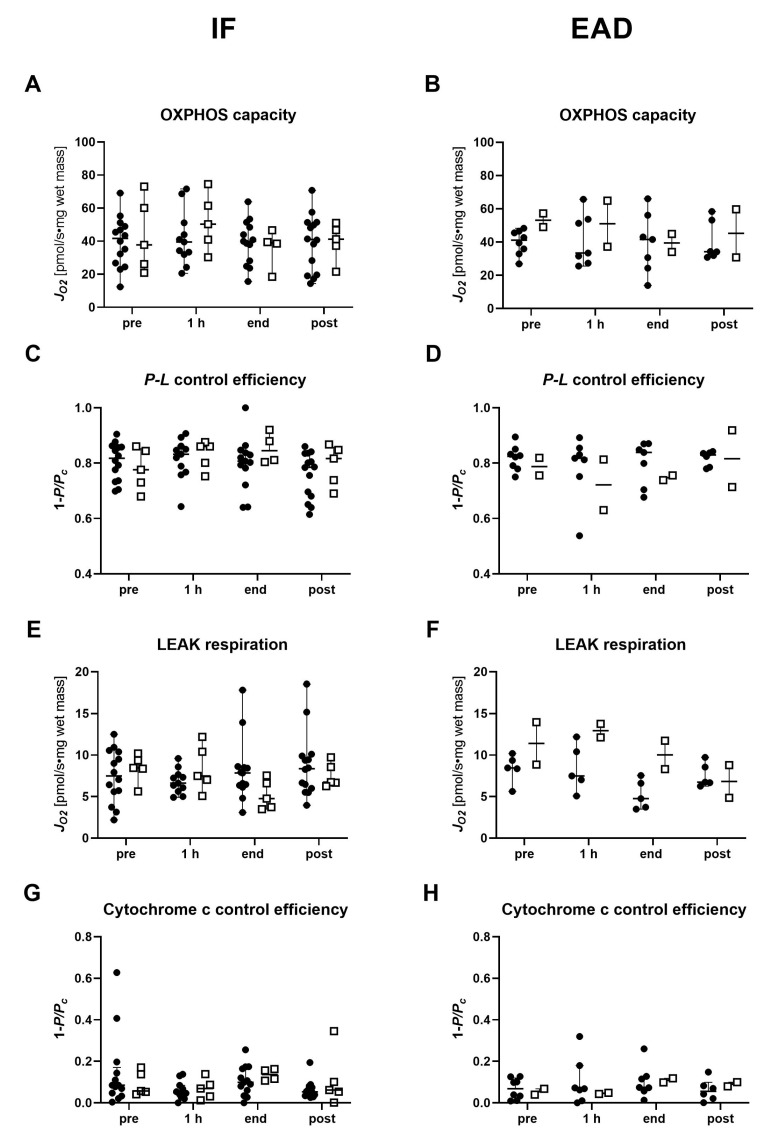
Analysis of succinate-linked mitochondrial respiration in tissue biopsies of transplanted DBD and DCD livers with initial function or early allograft dysfunction pre-NMP, at 1 h and at the end of NMP, and post-reperfusion. (**A**,**B**) OXPHOS capacity, (**C**,**D**) P-L control efficiency, (**E**,**F**) LEAK respiration and (**G**,**H**) cytochrome *c* control efficiency were measured and calculated following the protocol as described in Appendix A. Results are shown as individual values and medians for DBD (solid circles) and DCD (open boxes) livers. IF, initial function; EAD, early allograft dysfunction; NMP, normothermic machine perfusion; h, hour; DBD, donation after brain death; DCD, donation after circulatory death.

**Figure 8 ijms-24-09536-f008:**
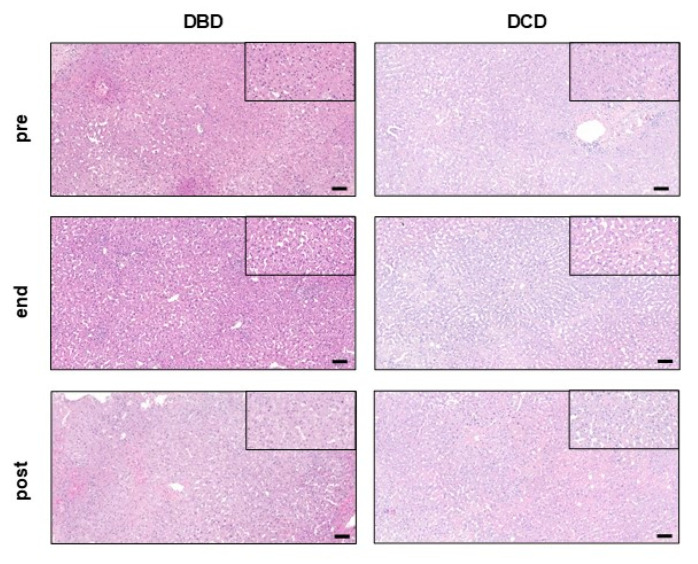
Histopathology of transplanted DBD and DCD livers pre-NMP, at the end of NMP and post-reperfusion. Representative images of haematoxylin & eosin stainings for DBD and DCD livers. Scale bar represents 100 µM (10× magnification). Black frame: 40× magnification. DBD, donation after brain death; DCD, donation after circulatory death; NMP, normothermic machine perfusion.

**Figure 9 ijms-24-09536-f009:**
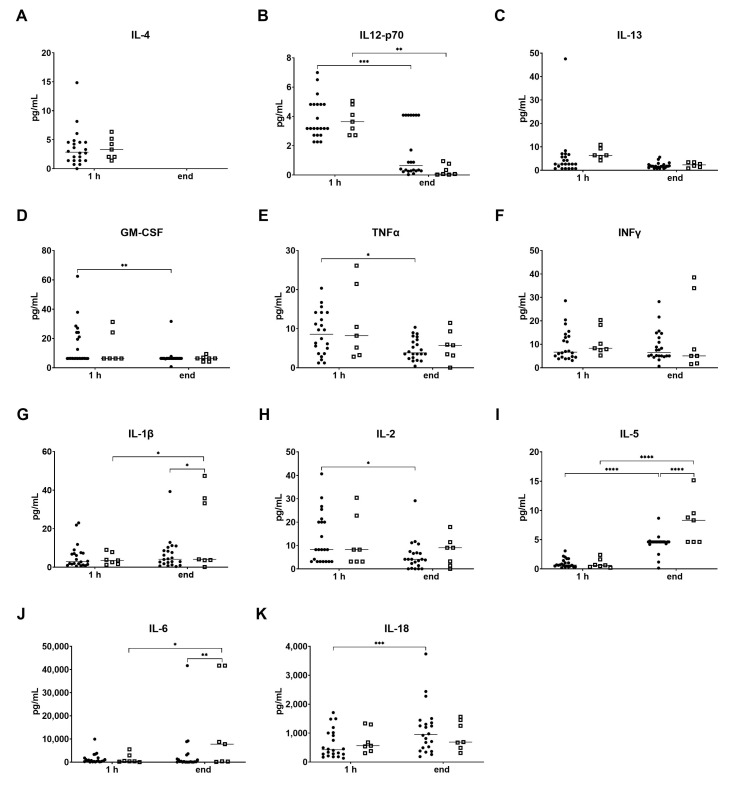
Perfusate cytokine levels at 1 h NMP and at the end of NMP of transplanted DBD and DCD livers. Anti-inflammatory cytokines (**A**) IL-4, (**B**) IL-12 p70, (**C**) IL-13, and pro-inflammatory cytokines (**D**) GM-CSF, (**E**)TNFα, (**F**) IFNγ, (**G**) IL-1β, (**H**) IL-2, (**I**) IL-5, (**J**) IL-6 and (**K**) IL-18 are shown as individual values and medians for DBD (solid circles) and DCD (open boxes) livers. (* *p* < 0.05, ** *p* < 0.01, *** *p* < 0.001, **** *p* < 0.0001; ANOVA tests between time points and between the groups; details are described in the section on statistical analysis). NMP, normothermic machine perfusion. h, hour; IL, Interleukin; GM-CSF, granulozyte-monozyte colony-stimulating factor; TNFα, tumor necrosis factor alpha; IFNγ, interferon gamma.

**Figure 10 ijms-24-09536-f010:**
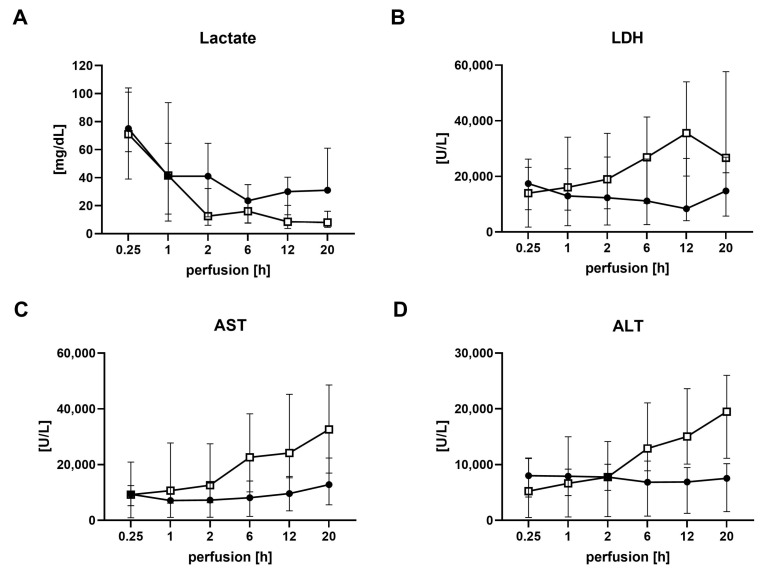
Analysis of perfusate parameters during normothermic machine perfusion of DBD and DCD livers finally declined for transplantation. (**A**) Lactate, (**B**) Lactate dehydrogenase (LDH), (**C**) Aspartate aminotransferase (AST) and (**D**) Alanine aminotransferase (ALT) for DBD donors (solid circles) and DCD donors (open boxes). Values are expressed as the median and interquartile range. h, hours; DBD, donation after brain death; DCD, donation after circulatory death.

**Figure 11 ijms-24-09536-f011:**
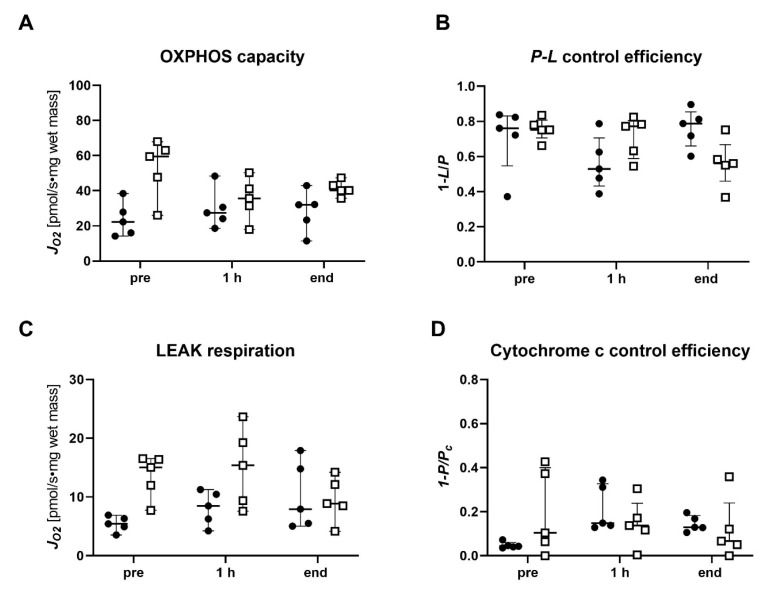
Analysis of succinate-linked mitochondrial respiration in tissue biopsies of declined DBD and DCD livers pre-NMP, at 1 h NMP and at the end of NMP. (**A**) OXPHOS capacity, (**B**) P-L control efficiency, (**C**) LEAK respiration and (**D**) cytochrome *c* control efficiency were measured and calculated following the protocol as described in Appendix A. Values are represented as single data points, medians and interquartile ranges for DBD (solid circles) and DCD (open boxes) livers. NMP, normothermic machine perfusion; h, hour; DBD, donation after brain death; DCD, donation after circulatory death.

**Table 1 ijms-24-09536-t001:** Donor demographics.

Donor Characteristics	Overall Cohort (*n =* 40)	DBD (*n =* 27)	DCD (*n =* 13)	*p* Value
Age [y, median (IQR)]	62 (51–70)	66 (53–75)	53 (48–59)	0.0075
Sex ratio [*n* (%)]				
Male	23 (57.5%)	12 (44.4%)	11 (84.6%)	
Female	17 (42.5%)	15 (55.6%)	2 (15.4%)	0.0204
BMI [kg/m^2^, median (IQR)]	28.0 (25.0–30.0)	28.0 (26.0–32.0)	26.0 (24.0–28.0)	0.1165
Cause of death				
CVA	22	19	3	
Circulatory	7	2 *	5 ^#^	
Trauma	8	5	3	
Other	3	1	2	
ECD [*n* (%)]	37 (92.5%)	24 (88.8%)	13 (100%)	0.6847
DRI [median (IQR)]	1.89 (1.72–2.32)	1.80 (1.65–1.80)	2.35 (2.06–2.88)	0.0007

* hypoxic brain damage (*n* = 1), not otherwise specified (*n* = 1); # cardiac arrest (*n* = 5); DBD, donation after brain death; DCD, donation after circulatory death; y, years; IQR, interquartile range; BMI, body mass index; CVA, cerebrovascular accident; ECD, extended criteria donor; DRI, donor risk index.

**Table 2 ijms-24-09536-t002:** Organ preservation and normothermic machine perfusion times.

Preservation and NMP	Overall Cohort (*n =* 40)	DBD (*n =* 27)	DCD (*n =* 13)	*p* Value
CIT [min, median (IQR)]	348 (300–409)	346.00 (285–406)	363 (305–482)	0.4036
WIT [min, median (IQR)]	25 (20.5–28.5)	-	25 (20.5–28.5)	
NMP [min, median (IQR)]	894 (708–1233)	809 (653–1177)	1106 (978–1294)	0.0061
Total preservation time [min, median (IQR)]	1206 (1037–1580)	1136 (1001–1530)	1523 (1255–1713)	0.0015
Accepted for LT [*n* (%)]	29 (72.5%)	22 (81.5%)	7 (53.9%)	

NMP, normothermic machine perfusion; DBD, donation after brain death; DCD, donation after circulatory death; CIT, cold ischemic time; IQR, interquartile range; min, minutes; WIT, warm ischemic time; LT, liver transplantation.

**Table 3 ijms-24-09536-t003:** Recipient demographics and clinical outcomes of transplanted livers.

Recipient Characteristics	Overall Cohort (*n =* 29)	DBD (*n =* 22)	DCD (*n =* 7)	*p* Value
Age [y, median (IQR)]	60 (50–68)	62 (47–70)	60 (56–63)	0.5909
Sex ratio [*n* (%)]				
Male	18 (62.1%)	14 (63.6%)	4 (57.1%)	
Female	11 (37.9%)	8 (36.4%)	3 (42.9%)	>0.9999
BMI [kg/m^2^, median (IQR)]	25.8 (20.9–29.6)	24.7 (20.6–29.7)	26.2 (23.8–28.6)	0.9901
MELD [median (IQR)]	12.0 (7.5–19.8)	12.0 (10.0–25.0)	12.0 (6.0–18.5)	0.2008
Clinical outcome				
L-GrAFT [median (IQR)]	−0.994 (−1.351–0.174)	−1.022 (−1.383–0.040)	−0.516 (−1.416–0.174)	0.6999
MEAF [median (IQR)]	4.835 (4.070–6.700)	4.905 (4.178–6.615)	4.680 (3.478–6.903)	0.7897
EAD	10 (34.5%)	8 (36.4%)	2 (28.6 %)	0.3298
1-year patient survival	25 (86.2%)	20 (90.9%)	5 (71.4%)	0.1930
1-year graft survival	25 (86.2%)	20 (90.9%)	5 (71.4%)	0.1930

DBD, donation after brain death; DCD, donation after circulatory death; y, years; IQR, interquartile range; BMI, body mass index; MELD, a model for end-stage liver disease; L-GrAFT, liver graft assessment following transplantation; MEAF, model for early allograft function; EAD, early allograft dysfunction.

**Table 4 ijms-24-09536-t004:** Semiquantitative assessment of histopathology in tissue biopsies of transplanted DBD and DCD livers pre-NMP, at the end of NMP and post-reperfusion.

	Overall Cohort (*n =* 29)	DBD (*n =* 22)	DCD (*n =* 7)	*p* Value
Necrosis [Median (IQR)]				
pre	0 (0–1)	0 (0–0)	0 (0–1)	0.9218
end	0 (0–1)	0 (0–1)	0 (0–1)	0.3116
post	0 (0–1)	0 (0–1)	0 (0–1)	0.9646
Steatosis [Median (IQR)]				
pre	0.5 (0–1)	1 (0 -1)	0 (0–1)	0.2908
end	0 (0–1)	0 (0–1)	0 (0–0)	0.2495
post	0 (0–1)	0 (0–1)	0 (0–0)	0.6098
Fibrosis [Median (IQR)]				
pre	0 (0–0)	0 (0–0)	0 (0–0)	0.9255
end	0 (0–0)	0 (0–0)	0 (0–0)	0.6096
post	0 (0–0)	0 (0–0)	0 (0–0)	0.9255
Inflammation [Median (IQR)]				
pre	1 (1–1)	1 (1–1)	1 (1–1)	0.9894
end	1 (1–1)	1 (1–1)	1 (1–1)	0.9247
post	1 (1–1)	1 (1–1)	1 (1–1)	0.9247
Vascular damage [Median (IQR])				
pre	1 (1–1)	0 (1–0)	1 (1–1)	0.1093
end	1 (1–1)	1 (1–1)	1 (1–1)	0.9969
post	1 (1–1)	1 (1–1)	1 (1–1)	0.7084

NMP, normothermic machine perfusion; DBD, donation after brain death; DCD, donation after circulatory death; IQR, interquartile range.

**Table 5 ijms-24-09536-t005:** Reasons for discarding non-transplanted DBD and DCD livers after normothermic machine perfusion.

DBD Liver	Reason for Discard
1	Inadequate lactate clearance, high perfusate transaminases
2	Inadequate lactate clearance and high perfusate IL-6 levels
3	Histology: macrosteatosis, physiological perfusate pH could not be maintained despite NaHCO_3_ addition, inadequate lactate clearance
4	Inadequate lactate clearance
5	Malignant tumor of the donor
**DCD Liver**	
1	High-perfusate transaminases
2	Inadequate lactate clearance
3	Technical problems related to NMP
4	Histology: fibrosis
5	High-risk organ (DCD); high perfusate transaminases
6	Arteriosclerosis Arteria hepatica, inadequate lactate clearance

DBD, donation after brain death; DCD, donation after circulatory death; IL, Interleukin; NaHCO_3_, sodium bicarbonate; NMP, normothermic machine perfusion.

**Table 6 ijms-24-09536-t006:** Perfusate cytokine levels of declined DBD and DCD livers at 1 h NMP and at the end of NMP.

Cytokine [Median (IQR)]	DBD (*n =* 5)	DCD (*n =* 6)	*p* Value
IL-4			
1 h	13.12 (2.04–25.03)	11.97 (7.20–13.12)	0.6087
end	<LOD	<LOD	
IL-12 p70			
1 h	6.26 (3.07–9.35)	5.30 (4.41–6.99)	0.7698
end	0.95 (0.54–2.34)	0.47 (0.27–1.29)	0.8650
IL-13			
1 h	7.07 (2.49–13.20)	6.40 (5.34–9.71)	0.9517
end	3.95 (2.31–6.08)	2.32 (1.25–3.46)	0.5671
GM-CSF			
1 h	25.60 (13.72–42.86)	18.59 (10.62–28.35)	0.0742
end	5.13 (3.91–5.13)	6.43 (4.64–7.92)	0.9334
TNFα			
1 h	12.68 (6.06–23.08)	11.20 (8.58–14.52)	0.7346
end	10.01 (7.69–16.67)	8.63 (4.66–11.10)	0.5032
IFNγ			
1 h	11.45 (6.54–27.20)	13.49 (9.84–20.56)	0.9996
end	23.04 (17.06–105.64)	9.01 (4.04–27.73)	0.1873
IL-1β			
1 h	9.01 (5.91–17.11)	5.08 (3.13–7.81)	0.8415
end	53.74 (16.39–78.18)	11.23 (5.22–28.02)	0.0118
IL-2			
1 h	21.44 (10.87–28.98)	15.50 (8.26–17.62)	0.4303
end	6.91 (6.50–10.45)	9.48 (6.01–25.13)	0.5223
IL-5			
1 h	3.37 (0.92–3.94)	1.85 (0.82–3.68)	0.9940
end	5.24 (4.78–7.97)	5.40 (4.23–14.55)	0.3620
IL-6			
1 h	1172.95 (243.10–2229.94)	458.2 (411.10–657.10)	0.9674
end	7829.01 (2559.54–18,138.94)	5938.39 (3702.15–13,128.94)	0.6633
IL-18			
1 h	408.91 (236.60–1384.93)	602.36 (393.23–1380.35)	0.9828
end	1147.04 (1097.29–3737.42)	655.43 (425.82–962.11)	0.0426

NMP, normothermic machine perfusion; LOD, limit of detection; IL, Interleukin; GM-CSF, granulozyte-monozyte colony-stimulating factor; TNFα, tumor necrosis factor alpha; IFNγ, interferon gamma; DBD, donation after brain death; DCD, donation after circulatory death; h, hour; IQR, interquartile range.

## Data Availability

The data presented in this study are available on request from the corresponding author.

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
