# Peer review of "Bioenergetic and Cytokine Profiling May Help to Rescue More DCD Livers for Transplantation"

_ijms, 2023, doi:10.3390/ijms24119536_

Round 1

Reviewer 1 Report

This paper from the Innsbruck group builds on previously published work looking at the bioenergetic status of livers during/after NMP. The authors compare a group of DBD vs DCD livers with respect to bioenergetics and cytokine release during and after NMP. I think overall this is an interesting paper with valuable contributions to the field. There are moderate revisions required to improve readability, consistency, and additional analyses that could improve significance.

1. Figures and figure legends are variable in quality and detail - no legend is provided for some figures making it difficult to determine which line is DCD vs. DBD for the casual reader. figure numbers are repeated and non-sequential. Numerous abbreviations in figure legends are not adequately defined.

2. I would recommend the authors define the HRR parameters in the methods or results section. The manuscript assumes that the reader is familiar with the authors' previous work and understands the HRR parameters, which most readers probably do not. It would be helpful for the average reader to just have the definitions of LEAK, P-L, OXPHOS, etc stated in the paper.

3. I think this is interesting work but I would suggest one further analysis that could potentially increase the value/significant of this study. The authors compare bioenergetic performance between DCD and DBD and show overall that they are similar in livers that were transplanted after NMP. However, I would like to see the bioenergetic performance in livers with adequate post-transplant function versus those with early allograft dysfunction (defined by L-graft, MEAF, or Olthoff - may need to test each case). This could bring real-world relevance to the study and may be able to help surgeons differentiate between livers at risk of post-transplant dysfunction using HRR during NMP for example (similar to reference 17).

4. The discussion paragraph lines 301-316 is irrelevant as this study reports on bioenergetics not the center's experience with NMP.

5. No limitations are discussed, though there are many - including small sample size, variable perfusion times (making the end-NMP biopsy uncontrolled), differences in donor demographics between DCD/DBD (for example older vs. young mitochondria/livers).

6. Did the authors measure perfusate FMN levels? This would be valuable for comparison with bioenergetic results 

7. Authors should comment on significance of succinate-dominated oxidative phosphorylation during NMP given that other groups (particularly the Zurich group) have demonstrated that a build up of succinate during SCS followed by NMP results in significant reactive oxygen species production (as compared to HOPE, where ROS are minimized). How do this study's results compare/contribute to the ongoing debate of NMP versus HOPE?

Overall good but grammar can be improved for readability

Reviewer 2 Report

This is potentially a very interesting manuscript. Some minor issues have been detected with how some of the information has been presented within the manuscript which require the attention of the authors. This includes the following-

1) Cardiac death is no longer the current terminology with respect to DCD organ donors. This term needs to be replaced throughout the manuscript by the term circulatory death

2) There needs to be some further information provided in the Introduction section as to why the allograft survivals are similar between DCD and DBD livers (although this has not been the case in all centers). This has a bit to do with donor and recipient selection practices. This pertains to the statement towards the end of the first paragraph in the Introduction section

3) It could be argued that NMP is not necessarily the ideal platform to assess the DCD donor liver now that in situ NRP is an option in some places. Hence the relevant statement in the second paragraph of the Introduction section is best altered.

4) Can the authors please expand on what are the Circulatory reasons for the cause of donor deaths (as per Table 1). Are these donors that have died of cerebral anoxia/hypoxia unrelated to cerebrovascular causes?

5) Can you please provide a precise definition of how the WIT was calculated in the Methods section along making it clearer as to what the CIT represents (is this the time between instillation of cold preservation fluid in the donor until the time that NMP was commenced)?

6) There are two Figure 1's in the manuscript. Can the authors please check the numbering of all of the Figures and ensure that they are all in the correct order and that there are no duplicate numbers.

7) Figure 3 requires a legend which depicts the information as to which line in each of the graphs represents the DCD livers and which line depicts the DBD livers

8) Is it correct that all of the relevant data in the results section for NMP pertains only to the liver allografts which were transplanted and not to those which were discarded? It is hard to tell at times looking at some of the data along with the accompanying text in the results section as to whether this is the case or not. This all needs to be made clearer in the Results section

Overall, there ae minimal concerns with respect to the quality of the english language in this particular manuscript

Round 2

Reviewer 1 Report

The authors have submitted a reasonably revised manuscript based on reviewer's comments. However, I found the manuscript even more difficult to read with numerous track changes/multiple color highlights/strikeout texts - I would suggest to the authors when submitting a revision that a simple revised manuscript be submitted with only new/revised text highlighted and all old text simply deleted, otherwise the reviewer struggles to put sentences together when reading through the innumerable track changes formatting. In addition, there are comments on the side of the manuscript in German (?) from one author to another, which is odd and inappropriate for a revised manuscript. This manuscript needs further formatting and revision before it can be adequately reviewed for publication.

Numerous grammatical errors throughout the text

Reviewer 2 Report

The manuscript now reads a lot better in light of the revisions which have been undertaken by the authors

Author Response

We are very grateful for the positive feedback and thank the reviewer again for the valuable suggestions.  With best regards